# Generalized Zero-Shot Learning with Deep Calibration Network

**Shichen Liu[†], Mingsheng Long[†](✉), Jianmin Wang[†], and Michael I. Jordan[♯]**
[†]School of Software, Tsinghua University, China
[†]KLiss, MOE; BNRist; Research Center for Big Data, Tsinghua University, China
[♯]University of California, Berkeley, Berkeley, USA
`liushichen95@gmail.com` `{mingsheng, jimwang}@tsinghua.edu.cn`
`jordan@berkeley.edu`

## Abstract

A technical challenge of deep learning is recognizing target classes without seen data. Zero-shot learning leverages semantic representations such as attributes or class prototypes to bridge source and target classes. Existing standard zero-shot learning methods may be prone to overfitting the seen data of source classes as they are blind to the semantic representations of target classes. In this paper, we study *generalized* zero-shot learning that assumes accessible to target classes for unseen data during training, and prediction on unseen data is made by searching on both source and target classes. We propose a novel Deep Calibration Network (DCN) approach towards this generalized zero-shot learning paradigm, which enables simultaneous *calibration* of deep networks on the confidence of source classes and uncertainty of target classes. Our approach maps visual features of images and semantic representations of class prototypes to a common embedding space such that the compatibility of seen data to both source and target classes are maximized. We show superior accuracy of our approach over the state of the art on benchmark datasets for generalized zero-shot learning, including AwA, CUB, SUN, and aPY.

## 1 Introduction

Remarkable advances in object recognition has been achieved in recent years with the prosperity of deep convolutional neural networks [7, 29, 46, 24]. Despite the exciting advances, most successful recognition models are based on supervised deep learning, which often requires large-scale labeled samples to learn category concept and visual representation [44]. This addiction of deep learning to large-scale labeled data has limited well-established deep models to tasks with only thousands of classes, while recognizing objects "in the wild" without labeled training data remains a major technical challenge to artificial intelligence. In many real applications, objects in different categories may follow a long-tailed distribution that some popular categories have a large number of training images while other categories have few or even no training images. Furthermore, collecting and annotating a large-scale set of representative examplar images for target categories is expensive and in many cases prohibitive [14]. In contrast to deep learning, humans can learn from few examples, by effectively transferring knowledge from relevant categories, and even recognize unseen objects [19]. Such capability of humans has motivated the active research in one-shot and zero-shot learning [8].

It is thus imperative to design versatile algorithms for zero-shot learning, which extends classifiers from source categories, of which labeled images are available during training, to target categories, of which labeled images are not accessible [32, 37]. Zero-shot learning (ZSL) has attracted wide attention in various research areas including face verification [30], object recognition [32], and video understanding [52]. The main idea of zero-shot learning is to recognize objects of target classes by

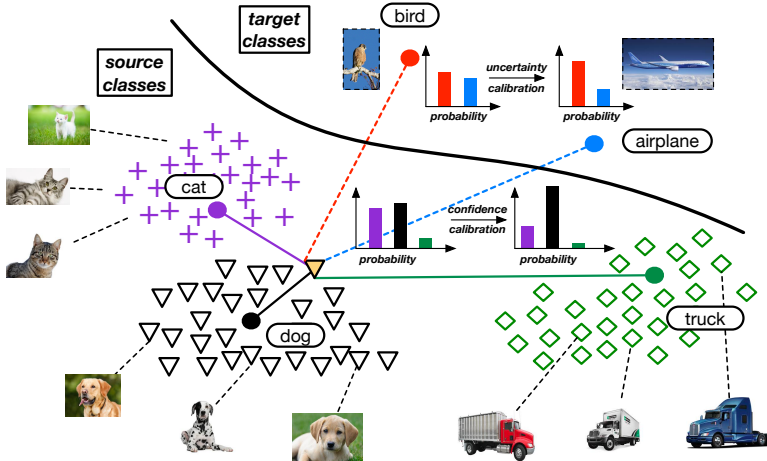

Figure 1: Technical difficulty for generalized zero-shot learning. Filled circles denote the semantic representations of all classes, empty markers denote seen data of source classes, and unseen data of target classes are unavailable. Given each training image, e.g. from a seen class *dog* denoted by the orange triangle, we can classify it to either source or target classes (the probability bars). However, overfitting the seen data to source classes will lead to uncertain predictions over target classes for either seen and unseen data, which potentially hurts the *generalized* zero-shot learning performance.

transferring knowledge from source classes through semantic representations of source and target classes, typically in the form of attributes [32] and class prototypes [8]. Towards this goal, we need to address two technical problems as envisioned in [8]: (i) how to relate target classes to source classes for knowledge transfer and (ii) how to make prediction on target classes without labeled training data.

Towards the first problem, visual attributes [12, 32, 38] and word embeddings [15, 35, 47] have been explored as semantic representations to correlate source and target classes. Most works leverage class embeddings directly as bridges between input images and output classes [1, 15, 36, 32, 42, 19], while others directly learn representations from class embeddings [17, 27, 60]. Towards the second problem, the probabilistic models [32] have been strong baselines for zero-shot learning. Classifiers for the target classes can be directly trained in the input feature space [1, 11, 34, 60] or can be trained in semantic space using the nearest prototype classifiers [15, 36, 17, 20, 19]. A recent method of synthesized classifiers (SynC) [8] exploited the existence of clustering structures in the semantic embedding space and constrained the two aligned manifolds of clusters corresponding to the semantic embeddings and the centers in the visual feature space. The work [9] further imposed the structural constraint that semantic representations must be predictive of the locations of their corresponding visual exemplars. These two representative approaches yielded state-of-the-art performance on ZSL.

Despite the encouraging advances in standard zero-shot learning where only source classes with seen data are available during training, their basic assumption that the unseen data only comes from target classes is not realistic for real applications, since unseen data may come naturally from both source and target classes. As a consequence, *generalized* zero-shot learning setting where prediction on unseen data is made over both source and target classes, has been proposed firstly in [10] and drawn increasing attention [45, 54]. Unfortunately, existing standard zero-shot learning methods may be prone to overfitting the seen data of source classes as they are blind to the semantic representations of target classes. Figure 1 intuitively demonstrates such a technical difficulty: as the seen data are overfit to the source classes, they are made distant away from the target classes, and the resultant model tend to make uncertain predictions for the unseen data over both source classes and target classes. Hence, how to extend deep networks to the generalized zero-shot learning setting remains an open problem.

Towards the above technical difficulty of generalized zero-shot learning, we propose a novel Deep Calibration Network (DCN) approach that enables simultaneous *calibration* of deep networks on the confidence of source classes and the uncertainty of target classes. The uncertainty of target classes is the main obstacle to generalized zero-shot learning, which is calibrated by the entropy minimization principle. The overconfidence of source classes is a side-effect of the high-capacity deep networks, which can be calibrated by the temperature distillation method [23]. We propose an end-to-end deep

architecture comprised of deep convolutional neural networks (CNN) and multilayer perceptrons (MLP) with new loss functions to enable end-to-end training by back-propagation, with which we map visual features of images and semantic representations of class prototypes to a common embedding space such that the compatibility of seen data to the target classes are maximized. We show superior accuracy of our approach over the state of the art on benchmark datasets (AwA, CUB, SUN, and aPY) for both generalized and standard zero-shot learning under rigorous evaluation protocols [54].

## 2 Related Work

Zero-shot learning models recognize target classes with no training samples through transferring knowledge from source classes with abundant training samples. Zero-shot learning was first studied by exploring visual attributes as semantic descriptions [12, 1, 16, 17, 30, 38, 32, 51, 4, 56]. This, however, requires manually-annotated attribute prototypes for each class, which is very costly for large-scale datasets. Recently, semantic word embeddings [35] are proposed to embed each class word by unsupervised learning from large-scale general text database, eliminating the need of human annotation of many visual attributes [47, 3, 15, 20, 36]. A combination of several embedding methods have also been investigated to improve the representation power for diverse category labels [3, 17, 20].

Based on semantic embeddings of class prototypes, existing zero-shot learning work can be generally categorized as embedding-based and similarity-based methods. In the embedding-based methods, one first maps visual features to the semantic space, and then predicts the class labels by various similarity measures implied by the class embeddings [1, 3, 15, 17, 20, 27, 32, 33, 36, 47, 51]. Some recent work combines these two stages for more accurate predictions [1, 3, 15, 42, 51, 59, 60]. In addition to the fixed semantic embeddings, some work maps them into a different space through subspace learning or feature learning [17, 27, 59, 61]. In the similarity-based methods, classifiers are built for target classes by relating them to the source classes through class-wise similarities [11, 20, 21, 34, 40, 41].

Towards the generalized zero-shot learning (GZSL) paradigm, Chao *et al.* [10] first studied this problem and tackled it with a calibrated stacking method to set different prediction thresholds for seen and unseen classes. Unlike setting thresholds for predictions, our method could learn a unified model through calibrating the uncertainty. Xian *et al.* [53] and Verma *et al.* [31] explored approaches that synthesize images for unseen classes using GANs [22] or VAE [26]. Despite their strong performance, these generative models are more difficult to train. Our approach significantly boosts the accuracy of GZSL in the regime of discriminative models, which attains the simplicity of model design. Our work is well complementary to the generative models based approaches. Another related setting is transductive zero-shot learning [18, 49] which exploits the unlabeled target data at training. While our approach does not require such data, it requires the availability of the target classes during training.

## 3 Generalized Zero-Shot Learning

In zero-shot learning, we are given seen data $\mathcal{D} = \{(\mathbf{x}_n, \mathbf{y}_n)\}_{n=1}^{N}$ of $N$ labeled points with labels from the *source* classes $\mathcal{S} = \{1, \ldots, S\}$, where $\mathbf{x}_n \in \mathbb{R}^P$ is the feature of the $n$-th image and $\mathbf{y}_n \in \mathcal{S}$ is the label. Denote by $\mathcal{T} = \{S+1, \ldots, S+T\}$ the *target* classes, where no seen data is available in the training phase. For each class $c \in \mathcal{S} \cup \mathcal{T}$, denote by $\mathbf{a}_c \in \mathbb{R}^Q$ its semantic representation (attributes or word embeddings), and by $\mathcal{A} = \{\mathbf{a}_c\}_{c=1}^{S+T}$ the set of semantic representations. In the test phase, we predict unseen data $\mathcal{D}' = \{\mathbf{x}_m\}_{m=N+1}^{N+M}$ of $M$ points from either source or target classes.

**Definition 1** (Zero-Shot Learning, ZSL). *Given $\mathcal{D}$ and $\{\mathbf{a}_c\}_{c=1}^{S}$, classify $\mathcal{D}'$ over target classes $\mathcal{T}$.*

**Definition 2** (Generalized Zero-Shot Learning, GZSL). *[10] Given $\mathcal{D}$ and $\{\mathbf{a}_c\}_{c=1}^{S+T}$ of both source and target classes, learn a model $f : \mathbf{x} \mapsto \mathbf{y}$ to classify $\mathcal{D}'$ over both source and target classes $\mathcal{S} \cup \mathcal{T}$.*

At first sight, it seems that the assumption of available semantic representation for the target classes $\{\mathbf{a}_c\}_{c=S+1}^{S+T}$ is too strong. But note that in *generalized* zero-shot learning we have to make predictions on unseen data over both source and target classes, and if we do not have access to the target classes, this generalized setting is highly undetermined and cannot be solved from a statistical perspective. Since we still hold the essential assumption of zero-shot learning that unseen data is not available in the training phase, as well as the semantic representations of target classes are not expensive to have, we believe this assumption is the mildest one we could have for real zero-shot learning applications. Figure 2 shows the proposed Deep Calibration Network (**DCN**) for generalized zero-shot learning.

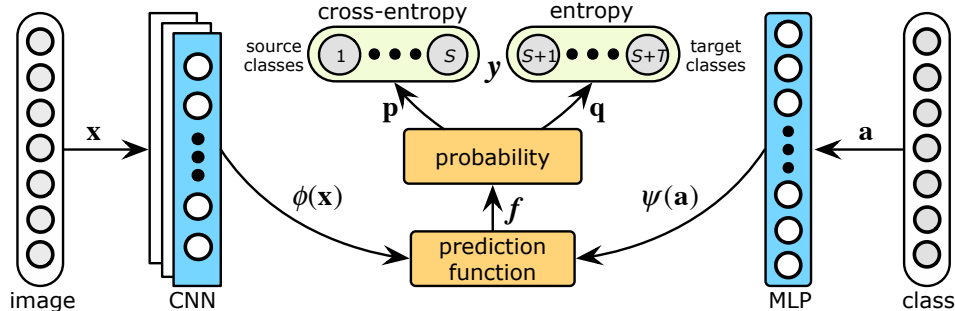

Figure 2: Deep Calibration Network (DCN) for generalized zero-shot learning, comprised of four modules: (i) a CNN for learning deep embedding $\phi(\mathbf{x})$ for each image $\mathbf{x}$ and an MLP for learning deep embedding $\psi(\mathbf{a})$ for each class $\mathbf{a}$; (ii) a prediction function $f$ made by nearest prototype classifier (NPC); (iii) two probabilities $p$ and $q$ that transform the prediction function $f$ into distributions over the source and target classes; (iv) a cross-entropy loss that minimizes the overconfidence of source classes, and an entropy loss that minimizes the uncertainty of target classes. *Best viewed in color.*

## 3.1 Prediction Function

The architecture of DCN consists of a visual model and a text model. We adopt deep convolutional networks as our visual models, e.g. GoogLeNet [48] and ResNet [24]. Deep convolutional networks can represent each image $\mathbf{x} \in \mathcal{D}$ by an feature embedding $\phi(\mathbf{x}) \in \mathbb{R}^K$, where $K$ is dimension of the embedding space. We adopt multilayer perceptrons (MLP) [43] as our text model, which can learn deep representation $\psi(\mathbf{a}) \in \mathbb{R}^K$ for each class $\mathbf{a} \in \mathcal{A}$ in the same $K$-dimensional embedding space. Each semantic class $\mathbf{a} \in \mathcal{A}$ can be represented by either word embeddings generated by Word2Vec [35], or by visual attributes annotated by humans to describe the visual patterns [32]. Note that, both images $\mathbf{x} \in \mathcal{D}$ and semantics $\mathbf{a} \in \mathcal{A}$ are mapped to the common $K$-dimensional embedding space as

$$\mathbf{z}_n = \phi(\mathbf{x}_n), \mathbf{v}_c = \psi(\mathbf{a}_c), \tag{1}$$

where $\mathbf{z}_n \in \mathbb{R}^K$ and $\mathbf{v}_c \in \mathbb{R}^K$ are visual embedding of images and semantic embedding of classes. Both networks utilize the nonlinearity function $\tanh$ that squash the predicted values within $[-1, 1]$.

Classification can be made in the shared embedding space by the nearest prototype classifier (NPC), which assigns to an image $\mathbf{x}_n$ the class label $c \in \mathcal{S} \cup \mathcal{T}$ whose semantic embedding $\psi(\mathbf{a}_c)$ is closest in similarity to the visual embedding $\phi(\mathbf{x}_n)$. Denote by $f$ the prediction function of the NPC classifier,

$$f_c(\mathbf{x}_n) = \text{sim}(\phi(\mathbf{x}_n), \psi(\mathbf{a}_c)), \tag{2}$$

where $\text{sim}(\cdot, \cdot)$ is a similarity function, e.g. inner product and cosine similarity; $f_c(\mathbf{x}_n)$ is the strength that NPC classifier assigns image $\mathbf{x}_n$ to class $c$. The $\tanh$ activation function further strengthens the nonlinearity of the nearest prototype classifier. The predicted class $y(\mathbf{x}_n)$ of image $\mathbf{x}_n$ is given by

$$y(\mathbf{x}_n) = \arg\max_c f_c(\mathbf{x}_n), \tag{3}$$

where prediction is made over both source and target classes as $c \in \mathcal{S} \cup \mathcal{T}$ in generalized zero-shot learning. Note that making predictions for different classes lead to different technical difficulties [54].

## 3.2 Risk Minimization

With the prediction $f(\mathbf{x}_n)$ and class labels $y_n$ in the seen data $\mathcal{D}$, one can perform classification by applying any well-established loss functions. We take the multi-class Hinge loss as an example:

$$\sum_{n=1}^{N} \sum_{c=1}^{S} \max(0, \Delta(y_n, c) + f_c(\mathbf{x}_n) - f_{y_n}(\mathbf{x}_n)), \tag{4}$$

where the margin is defined as $\Delta(y_n, c) = 0$ if $y_n = c$ and 1 otherwise. Most zero-shot learning methods (based on DeViSE [15]) use the multi-class Hinge loss to learn visual-semantic embeddings.

However, a very recent study [23] reveals that deep neural networks are no longer well-calibrated to the prediction confidence: the probability given by the Softmax outputs tends to make absolute

confident predictions, i.e. the maximum value of each Softmax output is close to 1, much higher than real confidences. Despite the higher accuracy of high-capacity deep models on many tasks, overfitting to the seen data of source classes may hurt transfer to target classes for generalized zero-shot learning.

Towards the above miscalibration of deep networks, we apply *temperature calibration* to mitigate the overconfidence to the source classes caused by overfitting the seen data. Temperature calibration was originally proposed by Hinton *et al.* [25] to distill knowledge from deep networks. We apply temperature calibration to transform prediction $f$ into probability distribution over source classes as

$$p_c\left(\mathbf{x}_n\right) = \frac{\exp\left(f_c\left(\mathbf{x}_n\right)/\tau\right)}{\sum\limits_{c'=1}^{S}\exp\left(f_{c'}\left(\mathbf{x}_n\right)/\tau\right)}, \qquad (5)$$

where $\tau$ is the temperature, and $\tau = 1$ is a common option in deep networks. Temperature $\tau$ "softens" the softmax (raises the output entropy) with $\tau > 1$. As $\tau \to \infty$, the probability $p_c \to 1/S$, which leads to maximum uncertainty. As $\tau \to 0$, the probability collapses to a point mass (i.e. $p_c = 1$). Since $\tau$ does not change the maximum of the softmax function, the class prediction remains unchanged if temperature calibration $\tau \neq 1$ is applied *after* convergence. However, note in this paper that the temperature calibration $\tau \neq 1$ is applied over the nearest prototype classifier (2) to encourage better-calibrated probabilities, which does avoid deep models from generating overconfident predictions. Unlike the distillation strategy [25, 23], we apply the temperature calibration $\tau \neq 1$ *during* training. Such a calibrated probability is essentially useful when we calibrate the uncertainty of target classes.

Plugging the probability $p_c$ (5) into cross-entropy loss over seen data $\mathcal{D}$ from source classes $\mathcal{S}$ yields

$$L = -\sum_{n=1}^{N}\sum_{c=1}^{S} y_{n,c}\log p_c\left(\mathbf{x}_n\right). \qquad (6)$$

It is worth noting that, comparing with the family of methods based on the multi-class Hinge loss in Eq. (4), this work differs in adopting the cross-entropy objective, which is a native solution to deal with multi-class problems and can take the power of temperature calibration to mitigate overfitting.

### 3.3 Uncertainty Calibration

As common practice of standard zero-shot learning, we can apply the trained model $f_c$ in Eq. (3) to classify the unseen data over only target classes $\mathcal{T}$. However, in *generalized* zero-shot learning, things become more difficult as we have to make predictions over both source and target classes $\mathcal{S} \cup \mathcal{T}$. Due to the huge gap between the disjoint source and target classes, a deep network trained on the source classes may still be under-confident for the target classes. In principle, this generalized zero-shot learning paradigm will be impossible without exploiting any structures from target classes.

In this paper, we enable generalized zero-shot learning by making our model unblind to target classes. It is important to note that the data associated with target classes are still inaccessible during training, so as to comply with the most important assumption of zero-shot learning. To remove the blindness, we first transform prediction $f_c$ into probability (with temperature calibration) over target classes as

$$q_c\left(\mathbf{x}_n\right) = \frac{\exp\left(f_c\left(\mathbf{x}_n\right)/\tau\right)}{\sum\limits_{c'=S+1}^{S+T}\exp\left(f_{c'}\left(\mathbf{x}_n\right)/\tau\right)}, \qquad (7)$$

Note that the temperature calibration $\tau \neq 1$ is applied *during* the end-to-end training of both Eq. (6) and Eq. (7). This strategy is enabled in the stochastic back-propagation procedure of deep networks.

Intuitively, probability $q_c$ should assign each seen image to the target classes as certain as possible, namely, classify it to the target classes that are most similar to the image's label in the source classes, rather than classify it to all target classes with equal uncertainty. In other words, we should minimize the uncertainty of classifying the seen data to target classes such that source classes and target classes are both made compatible with the seen data and thus comparable for generalized zero-shot learning. In information theory, entropy $h(q) = -q\log q$ is the basic measure of uncertainty of a distribution $q$. In this paper, we propose the objective for uncertainty calibration based on the entropy criterion as

$$H = -\sum_{n=1}^{N}\sum_{c=S+1}^{S+T} q_c\left(\mathbf{x}_n\right)\log q_c\left(\mathbf{x}_n\right). \qquad (8)$$

Note that making predictions of seen data over target classes as certain as possible does not imply making target classes as the predictions of seen data. Hence, the uncertainty calibration strategy will significantly improve prediction over target classes while having little harm on classifying seen data.

## 3.4 Deep Calibration Network

The optimization problem of the deep calibration network (**DCN**) for generalized zero-shot learning can be formulated by integrating the empirical risk minimization (6) and uncertainty calibration (8):

$$\min_{\phi,\psi} L + \lambda H + \gamma \Omega\left(\phi, \psi\right), \tag{9}$$

where $\Omega(\phi, \psi)$ is the penalty to control model complexity, and $\lambda$ and $\gamma$ are hyper-parameters. In deep learning, weight decay can be used to replace the penalty term $\gamma \Omega(\phi, \psi)$, thus we need not explicitly deal with this penalty term. After model convergence, we can apply the prediction function $f_c$ in Eq. (3) to make predictions on the unseen data over both source and target classes. The network parameters $\{\phi, \psi\}$ can be efficiently optimized by standard back-propagation with auto-differentiation technique supported in PyTorch[1]. It is worth noting that, by changing the deep architectures to other shallow methods (e.g. logistic regression), our approach can be readily applied to existing (generalized) zero-shot learning methods, provided that the outputs are probability distributions [5, 6].

# 4 Experiments

We perform extensive evaluation with state of the art methods for zero-shot and generalized zero-shot learning on four benchmark datasets, which will validate the efficacy of the proposed DCN approach.

## 4.1 Setup

**Datasets**    The statistics of the four benchmark datasets for zero-shot learning are shown in Table 1.

Animals with Attributes (**AwA**) [32] is a widely-used dataset for coarse-grained zero-shot learning, containing 30,475 images from 50 different animal classes with at least 92 labeled examples per class. A standard split into 40 source classes and 10 target classes is provided by the dataset creators [32].

Caltech-UCSD-Birds-200-2011 (**CUB**) [50] is a fine-grained dataset with large number of classes and attributes, containing 11,788 images from 200 different types of birds annotated with 312 attributes. The first zero-shot split of CUB with 150 source classes and 50 target classes was introduced in [2].

SUN Attribute (**SUN**) [39] is a fine-grained dataset, medium-scale in the number of images, containing 14,340 images from 717 types of scenes annotated with 102 attributes. We adopt the standard split of [32], containing 645 source classes (in which 65 classes are used for validation) and 72 target classes.

Attribute Pascal and Yahoo (**aPY**) [13] is a small-scale dataset with 64 attributes and 32 classes. We follow split in [54] and use 20 Pascal classes as source classes and 12 Yahoo classes as target classes.

| Dataset | # Attributes | # Source Classes | # Target Classes | # Images |
|---------|--------------|------------------|------------------|----------|
| SUN | 102 | 645 | 72 | 14,340 |
| CUB | 312 | 150 | 50 | 11,788 |
| AwA | 85 | 40 | 10 | 30,475 |
| aPY | 64 | 20 | 12 | 18,627 |

Table 1: Statistics of the four zero-shot learning datasets.

**Image features**    Due to variations in image features used by different zero-shot learning methods, we opt to a fair comparison with state of the art methods methods based on widely-used features: 2048-dimensional ResNet-101 features provided by [54] and 1024-dimensional GoogLeNet features provided by [8]. Classification accuracies of existing methods are directly reported from their papers.

**Class embeddings**    Class embeddings are important for zero-shot learning. As class embeddings for aPY, AwA, CUB and SUN, we use the per-class continuous attributes provided with the datasets. Note that the proposed method can also use the Word2Vec representations [35] as class embeddings.

**Comparison methods** We choose to compare with many competitive or representative methods, including shallow methods **DAP** [32], **ALE** [1], **SJE** [3], **ESZSL** [42], **ConSE** [36], **SynC** [8], **EXEM** [9], **ZSKL** [57], and deep methods **DeViSE** [15], **CMT** [47], **MTMDL** [55], **DEM** [58].

**Model variants** We further study two variants of the proposed **DCN** approach to justify the efficacy of the entropy and temperature calibration strategies respectively: (i) **DCN w/o E** is the variant of DCN without entropy calibration, i.e. setting $\lambda = 0$ in Equation (9). (ii) **DCN w/o ET** is the variant of DCN without entropy and temperature calibrations, i.e. setting $\lambda = 0$ and $\tau = 1$ in Equation (9).

**Protocols** Due to variations in the evaluation protocols for different methods, we conduct extensive experiments based on two typical protocols: **Standard Protocol** [32] and **Rigorous Protocol**[2] [54].

For benchmark datasets and standard splits, we follow exactly the **Standard Protocol** [32, 8] for the **AwA**, **CUB** and **SUN** datasets using GoogLeNet features as well as the **Standard Splits (SS)**, which enables a direct comparison with the published results. For **CUB** and **SUN**, we average the results of 4 and 10 splits provided by [8] respectively. We report the average per-image classification accuracy based on five random experiments, where the accuracy is computed over the images in target classes. For deep methods, we use GoogLeNet-V2 as base network on standard splits of **AwA** and **CUB** [58].

Recent study [54] has shown that Standard Protocol is not rigorous to benchmark zero-shot learning. A new Rigorous Protocol is proposed for three reasons: (a) The image features are ImageNet pre-trained ResNet-101 features, which have higher accuracy than GoogLeNet features, yielding more stable comparison across different methods; (b) The **Proposed Split (PS)** guarantee that no target classes are from ImageNet-1K since it is used to pre-train the base network, otherwise unfairness would be introduced; (c) The zero-shot performance is evaluated based on per-class (instead of per-image) classification accuracy Eq. (10), which accounts for the imbalances in the target classes. We thus adopt this **Rigorous Protocol** [54] for fair comparison on **AwA**, **CUB**, **SUN**, and **aPY** datasets.

$$\text{ACC}_{\mathcal{C}} = \frac{1}{|\mathcal{C}|} \sum_{c \in \mathcal{C}} \frac{\text{\#correctly predicted samples in class } c}{\text{\#samples in class } c}. \tag{10}$$

At test phase of zero-shot learning, test images are restricted to target classes and the search space is restricted to the target classes $\mathcal{C} = \mathcal{T}$. At the test phase of *generalized* zero-shot learning, the search space includes both source and target classes $\mathcal{C} = \mathcal{S} \cap \mathcal{T}$, a more practical but challenging setting.

We compute two accuracies: **ACC$_{ts}$**, accuracy of all unseen images in target classes; **ACC$_{tr}$**, accuracy of some seen images from source classes which are not used for training, as used by the **Proposed Split** in the **Rigorous Protocol** [54]. Then we compute the harmonic mean of the two accuracies as

$$\text{ACC}_{\text{H}} = \frac{2\text{ACC}_{ts} \times \text{ACC}_{tr}}{\text{ACC}_{ts} + \text{ACC}_{tr}}. \tag{11}$$

We choose harmonic mean as final criterion to favor high accuracies on both source and target classes.

**Implementation details** Our end-to-end trainable approach is implemented using PyTorch. We use a single-layer Multilayer Perceptron (MLP) to transform the attributes to the common embedding space without changing their dimensions; We add a FC layer at the top of CNNs (GoogLeNet-v2 & ResNet-101) to transform image representations to the same dimensions as the attributes. We train the FC layers and fine-tune the CNNs end-to-end by optimizing (9). We use stochastic gradient descent with 0.9 momentum and a mini-batch size of 64. We cross-validate the learning rate in $[10^{-4}, 10^{-1}]$, the temperature $\tau \in [0.1, 10]$, and the entropy-penalty parameter $\lambda \in [10^{-3}, 10^{-1}]$. To make comparison more readable and direct, we first discuss the results of standard zero-shot learning.

## 4.2 Results of Standard Zero-Shot Learning

**Standard Splits (SS)** The results on Standard Splits [32] of the three datasets are reported in Table 2. Sha *et al.* [8, 9] provided very comprehensive results on typical ZSL methods under rigorous evaluation protocols, we thus adopted their published results for direct comparison. We highlight the following results. (i) DCN significantly outperforms both shallow and deep zero-shot learning methods. This justifies the significance of the proposed cross-entropy loss (6) over the temperature-calibrated Softmax probability to train deep networks for zero-shot learning. (ii) DCN further

| Type | Method | AwA | CUB | SUN | Avg |
|------|--------|-----|-----|-----|-----|
| Shallow | DAP [32] | 60.5 | 39.1 | 44.5 | 48.0 |
| | ALE [1] | 53.8 | 40.8 | 53.8 | 49.5 |
| | SJE [3] | 66.3 | 46.5 | 56.1 | 56.3 |
| | ESZSL [42] | 59.6 | 44.0 | 8.7 | 37.4 |
| | ConSE [36] | 63.3 | 36.2 | 51.9 | 50.5 |
| | SynC [8] | 72.9 | 54.5 | 62.7 | 63.4 |
| | EXEM [9] | 76.5 | **58.5** | 67.3 | 67.4 |
| Deep | DeViSE* [15] | 56.7 | 33.5 | – | – |
| | CMT* [47] | 60.8 | 39.6 | – | – |
| | MTMDL* [55] | 63.7 | 32.3 | – | – |
| | DCN w/o ET | 82.0 | 52.3 | 66.4 | 66.9 |
| | DCN w/o E | **82.3** | 55.6 | **67.4** | **68.4** |

Table 2: Results of Standard Zero-Shot Learning on SUN, CUB, and AwA using Standard Split (SS) with GoogLeNet features [8]. *indicates the deep methods that are fine-tuned from GoogLeNet-v2.

| Type | Method | SUN | CUB | AwA | aPY | Avg |
|------|--------|-----|-----|-----|-----|-----|
| Shallow | DAP [32] | 39.9 | 40.0 | 44.1 | 33.8 | 39.5 |
| | ALE [1] | 58.1 | 54.9 | 59.9 | 39.7 | 53.2 |
| | SJE [3] | 53.7 | 53.9 | **65.6** | 32.9 | 51.5 |
| | ESZSL [42] | 54.5 | 53.9 | 58.2 | 38.3 | 51.2 |
| | SAE [28] | 40.3 | 33.3 | 53.0 | 8.3 | 33.7 |
| | ConSE [36] | 38.8 | 34.3 | 45.6 | 26.9 | 36.4 |
| | SynC [8] | 56.3 | 55.6 | 54.0 | 23.9 | 47.5 |
| Deep | DeViSE [15] | 56.5 | 52.0 | 54.2 | 39.8 | 50.6 |
| | CMT [47] | 39.9 | 34.6 | 39.5 | 28.0 | 35.5 |
| | DCN w/o ET | 59.0 | 53.2 | 63.4 | 42.0 | 54.4 |
| | DCN w/o E | **61.8** | **56.2** | 65.2 | **43.6** | **56.7** |

Table 3: Results of Standard Zero-Shot Learning using Proposed Split (PS) ResNet-101 features [54].

improves the accuracy from DCN w/o ET, the variant of DCN without using temperature calibration. This validates the effectiveness of temperature calibration for mitigating the deep networks from overfitting the seen data of source classes and thus improving the zero-shot generalization capability.

**Proposed Splits (PS)**  The results on Proposed Splits (PS) [32] of the four datasets are reported in Table 3, using ResNet-101 features and normalized attributes as class embeddings. DCN outperforms all previous methods substantially and consistently. (i) Our model performs better for datasets with relatively large number of classes, e.g. SUN and CUB. We conjecture that the proposed cross-entropy loss is a native solution to multi-class classification problems, which changes the loss design for zero-shot learning methods. (ii) More importantly, DCN outperforms DCN w/o ET, which validates that the temperature calibration can lead to consistently improved zero-shot classification accuracy.

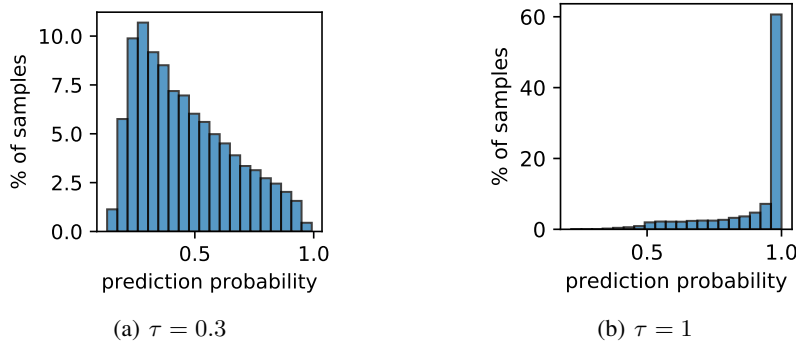

(a) $\tau = 0.3$  (b) $\tau = 1$

Figure 3: Probability histograms of DCN with ($\tau = 0.3$) or without ($\tau = 1$) temperature calibration.

**Result Analysis** We show in Figure 3 the histograms of probabilities output by our deep models with ($\tau = 0.3$) or without ($\tau = 1$) temperature calibration on AwA test data from the 10 target classes. As shown, DCN ($\tau = 1$) produces absolute probabilities for test data from the target classes, implying undesirable overconfidences for zero-shot learning. DCN ($\tau = 0.3$), the variant with temperature calibration, produces discriminative probabilities that is not prone to overfitting and overconfidences.

| Methods | SUN ts | SUN tr | SUN H | CUB ts | CUB tr | CUB H | AwA ts | AwA tr | AwA H | aPY ts | aPY tr | aPY H |
|---|---|---|---|---|---|---|---|---|---|---|---|---|
| DAP [32] | 4.2 | 25.1 | 7.2 | 1.7 | 67.9 | 3.3 | 0.0 | **88.7** | 0.0 | 4.8 | 78.3 | 9.0 |
| ALE [1] | 21.8 | 33.1 | 26.3 | 23.7 | 62.8 | 34.4 | 16.8 | 76.1 | 27.5 | 4.6 | 73.7 | 8.7 |
| SJE [3] | 14.7 | 30.5 | 19.8 | 23.5 | 59.2 | 33.6 | 11.3 | 74.6 | 19.6 | 3.7 | 55.7 | 6.9 |
| ESZSL [42] | 11.0 | 27.9 | 15.8 | 12.6 | 63.8 | 21.0 | 6.6 | 75.6 | 12.1 | 2.4 | 70.1 | 4.6 |
| ConSE [36] | 6.8 | 39.9 | 11.6 | 1.6 | **72.2** | 3.1 | 0.4 | 88.6 | 0.8 | 0.0 | **91.2** | 0.0 |
| SynC [8] | 7.9 | **43.3** | 13.4 | 11.5 | 70.9 | 19.8 | 8.9 | 87.3 | 16.2 | 7.4 | 66.3 | 13.3 |
| DeViSE [15] | 16.9 | 27.4 | 20.9 | 23.8 | 53.0 | 32.8 | 13.4 | 68.7 | 22.4 | 4.9 | 76.9 | 9.2 |
| CMT [47] | 8.1 | 21.8 | 11.8 | 7.2 | 49.8 | 12.6 | 0.9 | 87.6 | 1.8 | 1.4 | 85.2 | 2.8 |
| ZSKL [57] | 20.1 | 31.4 | 24.5 | 21.6 | 52.8 | 30.6 | 18.9 | 82.7 | 30.8 | 10.5 | 76.2 | 18.5 |
| DCN w/o ET | 23.8 | 36.1 | 28.7 | 26.5 | 52.5 | 35.2 | 17.2 | 84.7 | 28.6 | 8.3 | 73.0 | 15.0 |
| DCN w/o E | 23.9 | 37.2 | 29.1 | 25.9 | 65.8 | 37.2 | 17.2 | 84.7 | 28.6 | 8.4 | 72.2 | 15.1 |
| DCN | **25.5** | 37.0 | **30.2** | **28.4** | 60.7 | **38.7** | **25.5** | 84.2 | **39.1** | **14.2** | 75.0 | **23.9** |

Table 4: Results of Generalized Zero-Shot Learning on four datasets under Proposed Splits (PS) [54].

## 4.3 Results of Generalized Zero-Shot Learning

In real applications, whether an image is from a source or target class is unknown in advance. Hence, generalized zero-shot learning is a more practical and difficult task. Here, we use the same models trained on zero-shot learning setting on the Proposed Splits (PS) [54]. We evaluate our models on the generalized zero-shot learning setting, by evaluating performance on both source and target classes.

The results of generalized zero-shot learning are shown in Table 4, much lower than standard zero-shot learning. This is not surprising since the source classes are included in the search space which act as distractors for the images that come from target classes. As a result, methods performing better on source classes (e.g. DAP and ConSE) often perform worse on target classes, and vice versa (e.g. ALE and DeViSE). This reveals the indispensability of making models unblind to the target classes.

The DCN is to address this problem to yield much higher Harmonic accuracy $ACC_H$ and $ACC_{ts}$ on all datasets. By imposing a mild assumption that the semantic representation for target classes (cheap to have) is available during training, DCN is made accessible to target classes, which bridges the source and target classes through both semantic representations and the seen data. Note that DCN performs better DCN w/o E (without entropy penalty), highlighting the efficacy of the uncertainty calibration.

## 5 Conclusion

This paper proposed a deep calibration network towards generalized zero-shot learning. The approach enables simultaneous calibration of deep networks on the confidence of source classes and uncertainty of target classes, and as a consequence, bridges the source and target classes through both semantic representations of classes and visual embeddings of seen images. Experiments show that our approach yields state of the art performance for generalized zero-shot learning tasks on four benchmark datasets.

## Acknowledgments

This work was supported by the National Key R&D Program of China (2016YFB1000701), the Natural Science Foundation of China (61772299, 71690231, 61502265) and the DARPA Program on Lifelong Learning Machines.

## Footnotes

[1]`https://pytorch.org`

[2]http://www.mpi-inf.mpg.de/zsl-benchmark

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
