[Reviews · NeurIPS 2018]

Reviewer 1



UPDATE AFTER READING REVIEWS I have read the authors' rebuttal and my score remains unchanged. All reviewers seem to agree that the paper could use better clarity -- if the paper does get accepted, please consider a considerable effort towards this. ===== # Summary * The paper addresses generalized one shot learning. In this scenario, we know what the test classes will be (although we do not have acess to data from them). At test time, however, the data may also come from training classes. This is, generally speaking, harder than classical zero-shot learning. * The paper proposes using a neural network to jointly learn descriptors for the data and the classes, putting them in a common embedding space. * The main novelty is using a temperature in the softmax classifier, which intuitively controls the 'spikiness' of the confidence over classes. Together with an entropy term, this makes the network less over-confident of knowing seen classes, and gives larger probability to unseen classes. * Experiments on four datasets demonstrate competitive performance # Quality I think this paper has good quality. The ideas are simple -- albeit mildly well-explained. There is a clear motivation the evaluation is thorough. I think the paper could benefit from a more detailed section on related work. # Clarity I think this is a medium point in the paper. Although the concepts are simple, I find the sentences that try to explain it a bit confusing. Take for example this phrase: > Intuituively, probability $q_c$ should assign each seen image to the target classes as certain as possible, namely, classify it to the target classes that are most similar to the image’s label in the source classes, rather than classify it to all target classes with equal uncertainty. In other words, we should minimize the uncertainty of classifying the seen data to target classes such that source classes and target classes are both made compatible with the seen data and thus comparable for generalized zero-shot learning. The statements are bit too long for English, and contain multiple subordinate sentences each. I had to re-read this multiple times to understand the main point. # Originality I think this is a medium point of the paper. The contributions are mainly on the application of well known techniques to GZSL. In terms of application this is interesting, but not very original. # Significance I think this is a strong part of the paper. Although the modifications are simple (eg. temperature in softmax), they perform really well. This might be a new baseline that sets a new bar for future work in this area.

Reviewer 2



The paper focuses on generalized zero-shot learning, where prediction on unseen data is made over both source (visible during training) and target classes (not available during training). The proposed framework is called Deep Calibration Network (DCN). Related work is short but informative. The architecture of DCN consists of a visual model and a text model. An embedding space is filled with low-dimensional representation originated by both text (semantic representation) and images (visual representation). When an image is given, its visual representation is checked against the closest semantic representation, both projected in the embedded space. In this straightforward scheme, the novelty is in the temperature calibration, which acts as a sort of softmax regularizer, which mitigates the tendency toward to the source classes (intead of the unseen ones) caused by overfitting the seen data. This becomes tangible into the DCN in an explicit entropic term, together with a complexity penalty which adds to the standard loss. Experiments are consistent, convincing, detailed, exhaustive. In particular, and obviously, generalized tests make the wow effect.

Reviewer 3



POST REBUTTAL COMMENTS: I found the rebuttal effective, as it answered my main concerns. Therefore I upgraded my score from 6 to 7 as I believe the authors would improve the paper based on the rebuttal. I still think the title is overselling, and should be modified. Because the network architecture is standard, and the novelty is in the regularization and temperature calibration scheme. --------------------------------------------- The paper deals with zero-shot learning (ZSL). Which is about recognizing target classes with no training samples. Instead a semantic description is provided, which allows to transfer knowledge from source classes, to classify target classes. Specifically this paper focuses on improving a recently introduced “generalized” ZSL framework (Xian, CVPR 2017), where samples from both the source and target classes appear on test time. This framework is challenging, because during inference, the common ZSL models are mostly biased toward predicting source classes, rather than target classes. This paper proposes a novel idea to tackle this problem, arguing that in many cases the semantic description of the target classes is known during the training phase. Which can be used as additional signal during the training phase. To exploit this signal, the authors propose to use a network model that outputs a probability distribution (using a softmax). With that, they can calibrate the output distribution using two components: (1) Temperature calibration (Hinton 2015, Guo 2017) (2) A new regularization term for ZSL, which tries to minimize the entropy of target class prediction on training samples (of source classes). This term uses the target-class-semantic-description to produce the target class predictions. The regularization term is motivated by the conjecture that target classes may be similar to some source classes. The experiments show that using the proposed technique clearly improves the prediction accuracy, with respect to only having a probability distribution output. Comparing to other ZSL approaches show similar or improved accuracy. #Summary: I like the idea and the proposed regularization term, which are novel. But I gave a marginal score because quite a few details are vague or missing. Also, some important baselines are missing. I don’t think it should affect acceptance or rejection, because the performance of those baselines are published, and tested with the same datasets this paper uses. Yet, if the paper is accepted, it is important to relate to them. #Strengths: ##Novel ideas: (1) Take advantage of the fact that the semantic description of target classes is known during training. (2) The proposed entropy based regularization term, which appears to be effective. (3) Comparing to other ZSL approaches show similar or improved accuracy. #Weakness: ##The clarity of this paper is medium. Some important parts are vague or missing. 1) Temperature calibration: 1.a) It was not clear what is the procedure for temperature calibration. The paper only describes an equation, without mentioning how to apply it. Could the authors list the steps they took? 1.b) I had to read Guo 2017 to understand that T is optimized with respect to NLL on the validation set, and yet I am not sure the authors do the same. Is the temperature calibration is applied on the train set? The validation set (like Guo 2017)? The test set? 1.c) Guo clearly states that temperature calibration does not affect the prediction accuracy. This contradicts the results on Table 2 & 3, where DCN-T is worse than DCN. 1.d) About Eq (5) and Eq (7): Does it mean that we make temperature calibration twice? Once for source class, and another for target classes? 1.e) It is written that temperature calibration is performed after training. Does it mean that we first do a hyper-param grid search for those of the loss function, and afterward we search only for the temperature? If yes, does it means that this method can be applied to other already trained models, without need to retrain? 2) Uncertainty Calibration From one point of view it looks like temperature calibration is independent of uncertainty calibration, with the regularization term H. However in lines 155-160 it appears that they are both are required to do uncertainty calibration. (2.a) This is confusing because the training regularization term (H) requires temperature calibration, yet temperature calibration is applied after training. Could the authors clarify this point? (2.b) Regarding H: Reducing the entropy, makes the predictions more confident. This is against the paper motivation to calibrate the networks since they are already over confident (lines 133-136). 3) Do the authors do uncertainty calibration on the (not-generalized) ZSL experiments (Table 2&3)? If yes, could they share the ablation results for DCN:(T+E), DCN:T, DCN:E ? 4) Do the authors do temperature calibration on the generalized ZSL experiments (Table 4)? If yes, could they share the ablation results for DCN:(T+E), DCN:T, DCN:E ? 5) The network structure: 5.a) Do the authors take the CNN image features as is, or do they incorporate an additional embedding layer? 5.b) What is the MLP architecture for embedding the semantic information? (number of layers / dimension / etc..) ##The paper ignores recent baselines from CVPR 2018 and CVPR 2017 (CVPR 2018 accepted papers were announced on March, and were available online). These baseline methods performance superceed the accuracy introduced in this paper. Some can be considered complementary to this work, but the paper can’t simply ignore them. For example: Zhang, 2018: Zero-Shot Kernel Learning Xian, 2018: Feature Generating Networks for Zero-Shot Learning Arora, 2018: Generalized zero-shot learning via synthesized examples CVPR 2017: Zhang, 2017: Learning a Deep Embedding Model for Zero-Shot Learning ## Title/abstract/intro is overselling The authors state that they introduce a new deep calibration network architecture. However, their contributions are a novel regularization term, and a temperature calibration scheme that is applied after training. I wouldn’t consider a softmax layer as a novel network architecture. Alternatively, I would suggest emphasizing a different perspective: The approach in the paper can be considered as more general, and can be potentially applied to any ZSL framework that outputs a probability distribution. For example: Atzmon 2018: Probabilistic AND-OR Attribute Grouping for Zero-Shot Learning Ba 2015: Predicting Deep Zero-Shot Convolutional Neural Networks using Textual Descriptions Other comments: It will make the paper stronger if there was an analysis that provides support for the uncertainty calibration claims in the generalized ZSL case, which is the focus of this paper. Introduction could be improved: The intro only motivates why (G)ZSL is important, which is great for new audience, but there is no interesting information for ZSL community. It can be useful to describe the main ideas in the intro. Also, confidence vs uncertainty, were only defined on section 3, while it was used in the abstract / intro. This was confusing. Related work: It is worth to mention Transductive ZSL approaches, which use unlabeled test data during training, and then discriminate this work from the transductive setting. For example: Tsai, 2017: Learning robust visual-semantic embeddings. Fu 2015: Transductive Multi-view Zero-Shot Learning I couldn’t understand the meaning on lines 159, 160. Lines 174-179. Point is not clear. Sounds redundant. Fig 1 is not clear. I understand the motivation, but I couldn’t understand Fig 1.